# Digital Quantum Simulation of the Spin-Boson Model under Markovian Open-System Dynamics

**DOI:** 10.3390/e24121766

**Published:** 2022-12-02

**Authors:** Andreas Burger, Leong Chuan Kwek, Dario Poletti

**Affiliations:** 1Faculty of Physics, Ludwig-Maximilians-Universität Munich, Geschwister-Scholl-Platz 1, 80539 Munich, Germany; 2Science, Mathematics and Technology Cluster, Singapore University of Technology and Design, Singapore 487372, Singapore; 3Centre for Quantum Technologies, National University of Singapore, Singapore 117543, Singapore; 4National Institute of Education and Institute of Advanced Studies, Nanyang Technological University, Singapore 637616, Singapore; 5MajuLab, CNRS-UNS-NUS-NTU International Joint Research Unit, Singapore 117543, Singapore; 6EPD Pillar, Singapore University of Technology and Design, 8 Somapah Road, Singapore 487372, Singapore; 7The Abdus Salam International Centre for Theoretical Physics, Strada Costiera 11, 34151 Trieste, Italy

**Keywords:** quantum computing, NISQ, open system

## Abstract

Digital quantum computers have the potential to simulate complex quantum systems. The spin-boson model is one of such systems, used in disparate physical domains. Importantly, in a number of setups, the spin-boson model is open, i.e., the system is in contact with an external environment which can, for instance, cause the decay of the spin state. Here, we study how to simulate such open quantum dynamics in a digital quantum computer, for which we use an IBM hardware. We consider in particular how accurate different implementations of the evolution result as a function of the level of noise in the hardware and of the parameters of the open dynamics. For the regimes studied, we show that the key aspect is to simulate the unitary portion of the dynamics, while the dissipative part can lead to a more noise-resistant simulation. We consider both a single spin coupled to a harmonic oscillator, and also two spins coupled to the oscillator. In the latter case, we show that it is possible to simulate the emergence of correlations between the spins via the oscillator.

## 1. Introduction

A natural application of quantum computers is the simulation of quantum systems [1,2]. Furthermore, most hardware realizations of quantum computers implement the qubit. A prevalent qubit-based quantum system is the spin system. Existing quantum computers are based on unitary quantum circuits. Consequently, there has been a plethora of research on closed quantum systems [3,4,5,6]. Amongst the spin models, an important class is the spin-boson problem, where one or more spins are coupled to several bosonic degrees of freedom. These models possess rich many-body physics and they can model realistic coupling between electron transfer and protein motion or a solvent [7,8,9,10,11].

In recent years, NISQ computers [12,13] have offered a new perspective on the implementations on digital devices, leading to an explosion of activities. Not all computing tasks are amenable to quantum processing. Classical optimization can often perform better than quantum algorithms. The challenges of device-induced noise have led to the popularity of hybrid quantum-classical variational algorithms (VQA) that split the workload between a quantum and a classical processor. These techniques are ideally suited for the evaluation of different quantities such as eigenstates [14], general quantum approximate optimization algorithms [15], off-diagonal elements of matrices [16] and more [13]. Importantly, new error mitigation approaches have also been proposed [17,18,19]. VQA has been applied to boson-spin systems or its equivalents [20,21,22]. Regarding open systems, different VQA approaches have been tested. They include approaches based on imaginary time evolution [23,24], stochastic Schrödinger Equation [25], variational quantum eigensolvers to reach steady states [26,27], and the quantum-assisted simulator without a classical-quantum feedback loop [28]. Mapping bosonic problems to quantum circuits has been laid out in [29,30,31], while a recent implementation of spin-boson models can be found in [6].

Simulating open quantum systems entirely on digital quantum computers has primarily focused on two-level systems. The amplitude damping channel has been implemented with a unitary dilation of the Kraus operators [32], using uniformly controlled gates [33,34], and with the amplitude damping circuit [2,35]. Larger systems have been realized using a linear combination of unitary matrices [36,37] and modified stochastic Schrödinger equation methods [38]. In [25], the authors proposed a hybrid classical-quantum variational approach to simulate generic Markovian open quantum systems.

Our aim is to simulate the open dynamics of a spin-boson model coupled to a dissipative channel on a digital quantum computer. We do this by mapping the bosonic modes to qubits, Trotterizing the unitary evolution, and modeling the dissipative portion via repeated collisions with a reset auxiliary qubit [35,39,40]. In doing so, we focus on using different noise levels in the quantum computer, from the value in current hardware, to 1% of it. With this in mind, we study how different implementations of the simulation perform in presence of different noise levels.

The paper is organized as follows. In Section 2.1, we introduce the open spin-boson model and lay out the circuit implementation. In Section 2.2, we describe the circuit implementation of the unitary and dissipative evolutions. We then detail our use of quantum hardware and noise-related limitations of the devices in Section 2.3. Our results are presented in Section 3. We quantify the error stemming from approximations in the model, and for different magnitudes of noise in the device. We study the optimal time-step-sizes and dissipative rates in terms of fidelity. Finally, we increase the system size to two spins and investigate whether it is possible to observe rising correlations amongst the spins.

## 2. Method

### 2.1. Model

We consider $N_{S}$ non-interacting spins coupled to a single harmonic oscillator, as well as to a bath, as can be seen in Figure 1. The closed system is governed by the quantum Rabi Hamiltonian [41,42,43], which describes the ultra-strong coupling regime, where the usual rotating wave approximation breaks down and the counter-rotating term can no longer be neglected [44,45,46].
(1)$${\hat{H}}_{SB}=\hbar \omega {\hat{a}}^{†}\hat{a}+\sum_{i=1}^{N_{S}}\frac{1}{2}\left(h{\hat{\sigma }}_{k}^{z}+\epsilon {\hat{\sigma }}_{k}^{x}\right)+\lambda {\hat{\sigma }}_{k}^{x}\;\left({\hat{a}}^{†}+\hat{a}\right),$$
Experimentally, the ultra-strong coupling regime has been investigated in circuit QED [47,48,49,50,51,52], trapped ions [53], photonic systems [54], and semiconductors [55,56].

In Equation (Equation 1), ${\hat{a}}^{†}$ and $\hat{a}$, respectively, create and destroy one excitation in the harmonic oscillator while ${\hat{\sigma }}_{k}^{x}={\hat{\sigma }}_{k}^{+}+{\hat{\sigma }}_{k}^{-}$ and ${\hat{\sigma }}_{k}^{z}$ are Pauli operators acting on the spin(s). *h* is the local magnetic field in the *z* direction while ϵ is a field in the *x* direction. λ is the magnitude of the coupling between the spins and the harmonic oscillator, with frequency ω. In the following, we will work in units such that h=ℏ=1.

The dissipative part of the dynamics is here described by a Markovian master equation in Gorini–Kossakovski–Sudarshan–Lindblad form [57,58]
(2)$$\begin{array}{cc} \frac{d\hat{\rho }}{dt} & =-\frac{i}{\hbar }\left[{\hat{H}}_{SB},\hat{\rho }\right]+\gamma \sum_{k}\left(2{\hat{L}}_{k}\hat{\rho }{\hat{L}}_{k}^{†}-\left\{{\hat{L}}_{k}^{†}{\hat{L}}_{k},\hat{\rho }\right\}\right) \end{array}$$
with the amplitude damping channel ${\hat{L}}_{k}={|↓\rangle }_{k}\langle ↑|$ acting on the k−th spin and γ being the decay rate. |↓〉 represents the vacuum state, whereas |↑〉 represents the excited state of the spin.

Equation (Equation 2) describes a setup where loss from imperfections in the cavity is negligible compared to the spins emissions. In these systems, undesired decay transitions can include the emission of frequencies which are suppressed in the cavity and are thus effectively lost [59,60,61].

### 2.2. Circuit Implementation

In this section, we describe how we implement the evolution governed by Equations (Equation 1) and (Equation 2) in a quantum circuit.

#### 2.2.1. Encoding of the Hamiltonian

We map the spin and bosonic operators in ${\hat{H}}_{SB}$ to Pauli operators, and Trotterize the unitary $e^{-i{\hat{H}}_{SB}t}$. The spin part is trivially mapped to qubits. For the bosonic subspace and operators, we use a d-level-to-qubit mapping with Gray Code as the integer-to-bit encoding, as described in [31,62]. We give more details on the mapping to $Q_{B}$ qubits in Appendix A.

#### 2.2.2. Trotterization of Unitary

To implement the unitary evolution operator $U=e^{-i{\hat{H}}_{SB}t}$, we consider the first-order $U_{1}$ and second-order $U_{2}$ Suzuki–Trotter product formulas [63,64]
(3)$$\begin{array}{cc} \; & U_{1}={\left(e^{-ih_{1}\Delta t}\;e^{-ih_{2}\Delta t}\;\ldots \;e^{-ih_{N}\Delta t}\right)}^{\frac{t}{\Delta t}} \end{array}$$
(4)$$\begin{array}{c} U_{2}={\left(e^{-ih_{1}\frac{\Delta t}{2}}\;\ldots \;e^{-ih_{N}\frac{\Delta t}{2}}\;e^{-ih_{N}\frac{\Delta t}{2}}\;\ldots \;e^{-ih_{1}\frac{\Delta t}{2}}\right)}^{\frac{t}{\Delta t}} \end{array}$$
where $h_{k}$ are *N* different, non-commuting, terms of the Hamiltonian after encoding and Δt=t/N. The individual exponentials of Pauli strings $e^{-ih_{k}\Delta t}$ are then implemented via the CNOT-staircase [2,3], which is taken care of by Qiskit [65]. See Equations (Equation 11) and (Equation 12) in Appendix A for more details on $h_{k}$.

#### 2.2.3. Collisional Model

We model the local master equation, namely Equation (Equation 2), via repeated collisions [39,66]. Figure 2 gives a depiction of a single collision. We consider the spin qubit *s*, and auxiliary qubit *a* and where a controlled-$R_{Y}\left(\theta \right)$ (rotation around y axis) is followed by a controlled-NOT and a reset of the auxiliary qubit, see Appendix C and Appendix D for more details. To reproduce Equation (Equation 2) we use $\theta =\arcsin\left(\sqrt{1-e^{-\gamma t}}\right)$ [2].

#### 2.2.4. Integration of Dissipative and Unitary Part

To integrate the step of Figure 2 in the main circuit, we employ a first-order Suzuki–Trotter decomposition which alternates between the unitary and the dissipative parts. In Figure 3a, we depict three steps of the evolution of a single spin coupled to a harmonic oscillator mapped to two qubits, while in Figure 3b, we show our implementation of three-step evolution of the case with two spins and one harmonic oscillator. For considerations of connectivity, the auxiliary qubits needed for the dissipative channel are placed at the edges of the circuit, next to the spins. After all time-steps are finished, the qubits representing the spin(s) $s_{k}$ and the bosons $b_{k}$ are measured, while the state of the auxiliary qubit is ignored.

### 2.3. Quantum Hardware Simulation

To perform our quantum circuit simulations and run it on actual quantum hardware, we use IBM’s Qiskit software [65]. The Quantum Computer we use is the 7-qubit ibmq_jakarta device with a native gate set {CNOT, ID, RZ, SX, X}. Each circuit is run with $2^{13}=8192$ shots (repetitions).

We quantify the error at each point in time as the infidelity I [67]
(5)$$I\left(\hat{\rho },{\hat{\rho }}^{\prime }\right)=1-{\left(\mathrm{Tr}\left[\sqrt{\sqrt{\hat{\rho }}{\hat{\rho }}^{\prime }\sqrt{\hat{\rho }}}\right]\right)}^{2}$$
where we obtain the density matrix ${\hat{\rho }}^{\prime }$ of the circuit via quantum state tomography. We also consider a time-averaged version of the infidelity $\bar{I}$, which is obtained by averaging the infidelity over time, except for the time t=0, which consists of just the state preparation. The exact density matrix $\hat{\rho }$ for the benchmark is obtained from the exact evolution of the master Equation (Equation (Equation 2)), for which we use QuTiP [68]. To mitigate the measurement error on noisy hardware, we classically post-process the results with Qiskit’s error mitigation, which approximates the inverse of the noise matrix of the readout [69].

#### Reduced-Noise Models

While it is important to study how current quantum processors can evaluate the model we study, we also aim to explore what could be the performance of future, less noisy, hardware. To model these scenarios, we use the same error channels that IBM uses to describe their current devices.

The noise models include error sources in the gates, as thermal relaxation (relaxation and dephasing) and depolarizing errors, and also readout errors [70].

For our reduced-noise models we scale down the average gate infidelity $I_{Gate}$, the gate times $t_{Gate}$, and the false-readout probabilities, probability of measuring 1 when the state is 0 P(1|0) or vice versa P(0|1), by the same noise-factor ξ, or more precisely
(6)$$\begin{array}{cc} I_{Gate} & \to \xi \cdot I_{Gate} \end{array}$$
(7)$$\begin{array}{cc} t_{Gate} & \to \xi \cdot t_{Gate} \end{array}$$
(8)$$\begin{array}{cc} P(1|0),\;P(0|1) & \to \xi \cdot P(1|0),\;\xi \cdot P(0|1) \end{array}$$
where ξ ranges from 0 to 1.

Indeed, realistically some of these parameters will not see equal improvement in the coming years, but a more detailed analysis of differentiated improvements of different aspects is beyond the scope of this work. Details on the error channels can be found in Appendix B.

## 3. Results

Inaccuracies in the implementation of the model on a quantum computer can stem from different causes of completely different nature. We will first consider errors that rise from the Trotterization of the evolution in Section 3.1. We will then consider errors due to the noisy nature of the quantum computer in Section 3.2. In Section 3.3, we will then study the case of two spins coupled to the harmonic oscillator.

In the following, for the Hamiltonian, we choose the parameters ϵ=0.5, ω=4, λ=2 for one spin and ϵ=0.5, ω=6, λ=2 for two spins. For the open dissipative rate, we choose γ=1. With these parameters, an accurate evolution of the system up to a time t=2 can be obtained considering simply four levels for the harmonic oscillator, which can then be encoded with two qubits. For the initial state, we consider a pure product state between spins and bosons, with one spin in the excited state and zero excitations in the harmonic oscillator. This choice of initial conditions allows observing oscillatory, non-trivial dynamics from early times, while not requiring too many levels for the harmonic oscillator.

### 3.1. Error from the Circuit Implementation

As explained earlier, to implement the open dynamics, we Trotterize the unitary and dissipative parts of the master equation. However, for the implementation of the unitary evolution, we need to rely on another layer of Trotterization. In Figure 4, we consider a unitary evolution with Hamiltonian ${\hat{H}}_{SB}$ from Equation (Equation 1) for a time-step Δt and the possible implementation error, but considering no noise from the machine (blue lines). Implementing the various non-commuting terms of ${\hat{H}}_{SB}$ in Qiskit [65] requires 48 single-qubit- and 19 CX-Gates or 79 single-qubit- and 28 CX-Gates, when using first or second-order Trotter, respectively, (Table A1).

In Figure 4, we evaluate the infidelity for both unitary and dissipative evolutions, i.e., following Equation (Equation 2) for γ=0 (blue lines with circles) or γ=1 (orange lines with triangles), versus Δt. We observe that the second-order Trotterization, dashed lines, has significantly smaller infidelity than a first-order implementation, continuous lines. Interestingly, beyond Δt≈0.3, the infidelity in just the Hamiltonian simulation is larger than the infidelity when including the dissipation. Furthermore, independently of whether one considers first-order or second-order Trotterization, the dissipative dynamics has either smaller infidelity or it is very close to the unitary case. This implies that the unitary step implementing the Hamiltonian is the main contribution to the infidelity compared to the implementation of the dissipation.

### 3.2. Error in the Presence of Noise

We now turn to more realistic, and thus noisy, devices. In Figure 4, for noiseless simulations, we observed that the infidelity increases monotonously with the time-step size Δt, and that a second-order Trotterization is always preferred. In the presence of noise, however, an increased number of gates can lead to stronger noise effects, and thus instead of improving the quality of the simulations, it may result in worse fidelity. In Figure 5a, we thus consider the evolution of the full model, unitary and dissipative part, up to a time t=2 for different magnitudes of noise ξ=0.01,0.1,1 (from lighter to darker colors), for either a first-order Trotter step (continuous lines) or a second-order Trotter step (dashed lines). In particular, we depict the infidelity versus the length of the time-step Δt. We observe that for intermediate values of noise ξ=0.1,1 there is an optimal time interval Δt that corresponds to the lowest infidelity, and that first-order Trotterization can perform better at smaller Δt.

We now consider the open-system dynamics case. The impact of noise on fidelity is depicted in Figure 5b. Here, we show both the average infidelity over the time interval from t=0 to t=2 (blue line with circles), and the infidelity at the final time (orange line with triangles). We consider exclusively a second-order Trotter decomposition and a time-step Δt=0.2. Figure 5b indicates a monotonous growth of infidelity with the noise-factor ξ, for the parameters explored.

In Figure 6, we show the infidelity versus time for first-order (solid lines) and second-order (dashed line) Trotterizations, while Δt=0.2. We observe that, only for small values of ξ, one would prefer a second-order Trotterization to improve on the fidelity of the states. We note, not shown here, that for ξ=0.01, the dynamics is almost identical to the noiseless case.

To better understand the role of dissipation, we aim to verify its effect on the accuracy of the simulation. To focus specifically on the role of γ, we consider only a second-order Trotter evolution, a fixed value of Δt=0.2 and ξ=0.01, where the simulation of the quantum computer shows generally better performance compared to levels of higher magnitudes of noise ξ=0.1,1. In Figure 7a, we plot the time-averaged infidelity at different values of γ, with (an orange line with circles) and without noise (blue line with triangles). In noiseless simulations, the infidelity increases with γ, while in noisy simulations the infidelity initially reduces to a minimum at γ=1. Our understanding is that the dissipation in the exact calculations acts in a similar way as the intrinsic noise on the device, by drawing the system to its ground state and reducing coherence. It can thus be easier for a lossy quantum hardware to simulate a lossy system compared to a closed system (γ=0). However, a system with larger γ also implies further difficulties in the simulations stemming, for example, from Trotterization. It thus occurs that the intrinsic dissipative dynamics can, in some regimes, be better represented on a noisy device.

In Figure 7b, we plot the infidelity versus time for different values of the dissipative rate γ. We observe that, for γ≤1, the infidelity tends to increase with time, while for larger values of γ≥1.5, the infidelity can decrease after a maximum at an earlier time t≈0.4.

Figure 8 shows the average occupation in the harmonic oscillator, panel (a), and the expectation values of ${\hat{\sigma }}^{z}$ of the spin, panel (b), versus time. In both panels, the dotted line corresponds to the exact values, whilst the solid and dashed lines to ξ=0.01,0.1,1, respectively, from lighter to darker shades, and solid lines are used for first-order Trotterizations, while dashed lines are used for second-order. For each noise level ξ, we used the Trotterization order which corresponds to the lower fidelity.

The oscillatory evolution of the occupation of the harmonic oscillator is captured, only partially, with the smaller non-zero noise parameter considered ξ=0.01, panel (a), while the occupation of the harmonic oscillator at ξ=1 quickly stagnates around a value of 1. Instead, the simpler evolution of ${\hat{\sigma }}^{z}$ is also captured fairly well for the different values of ξ, as the simulated dissipation of the spin is closer to the relaxation of the spin-qubit under noise.

### 3.3. Two-Spin System

We here extend the system to two spins to see whether it is possible to the study correlation developing between them through a mediated interaction via the harmonic oscillator, as the two spins do not directly interact with each other. We use the parameters ϵ=0.5, ω=6, λ=2 and γ=1. We prepare the initial state in a product state of one spin in the excited state, one in the ground state, and the harmonic oscillator as completely empty. This allows us to observe non-trivial dynamics while still requiring just a few occupied levels of the harmonic oscillator.

As for the single-spin simulations, we first evaluate infidelity in the presence of noise. Simulating two spins requires roughly twice the number of gates as simulating one spin. A single Δt evolution with a first-order Trotter requires 113 single-qubits and 36 CX-Gates, while the second-order Trotter requires 177 single-qubits and 70 CX gates, as can be seen in Figure A1 and Table A1 in Appendix A. Furthermore, in the case of two spins, we find the optimal Trotter time-step Δt to be the same as for the single spin case (not shown).

To study the emerging correlations between the spins mediated by interaction with the photons, we consider the spin–spin correlators
(9)$$\begin{array}{c} C^{ZZ}=\langle {\hat{\sigma }}_{1}^{z}{\hat{\sigma }}_{2}^{z}\rangle -\langle {\hat{\sigma }}_{1}^{z}\rangle \langle {\hat{\sigma }}_{2}^{z}\rangle \\ C^{XX}=\langle {\hat{\sigma }}_{1}^{x}{\hat{\sigma }}_{2}^{x}\rangle -\langle {\hat{\sigma }}_{1}^{x}\rangle \langle {\hat{\sigma }}_{2}^{x}\rangle .\; \end{array}$$
These connected correlation functions (also called second-order Ursell functions or cumulants) correspond to the covariance in statistics and vanish if and only if ${\hat{\sigma }}_{1}^{\left(\cdot \right)}$ and ${\hat{\sigma }}_{2}^{\left(\cdot \right)}$ are statistically independent [71,72,73].

In Figure 9, we show $C^{ZZ}$ and $C^{XX}$ for, again, ξ=0.01,0.1,1 from lighter to darker lines. The solid lines correspond to first-order Trotter and dashed lines to second-order Trotter and these Trotterization orders have been chosen as they result, for the respective amount of noise, in the lowest infidelity. In both panels, the dotted lines correspond to the exact values. The exact case simulations show a build-up in anti-correlation in *z* direction at t=0.4, before reducing to 0 which can already be observed for ξ=0.1. A correlation in *x* direction builds up monotonously over time and one would need ξ=0.01 for a clearer signal.

In principle, correlations could be observed for a higher number of spins. In practice, the larger number of qubits needed, and their connectivity, would result in an increased number of gates which would limit the fidelity in NISQ devices. We also note that, going from one to two spins, we had to increase ω to keep the higher levels of the harmonic oscillator sparsely populated. If one does not want to increase the number of levels studied for the harmonic oscillator, a similar adjustment, such as decreasing the coupling between the harmonic oscillator and the spins, would be necessary when increasing the number of spins.

## 4. Conclusions

In this paper, we have studied the feasibility of simulating open spin-boson dynamics on a quantum computer. We used a second-quantization mapping of the bosonic degrees of freedom and Trotterization of the unitary to implement the Hamiltonian. To implement the dissipative dynamics, we used collisions and resets with auxiliary qubits.

We found that, in our parameter regime, the Hamiltonian simulation is the limiting factor to the fidelity. We surveyed optimal Trotterization formulas and time-step sizes depending on the level of noise in the system. We selected the open dissipative rate with the highest fidelity in noisy circuits, and we found that current noise levels in the machine we considered would make such simulations particularly challenging.

Anticipating future improved devices, we ran our simulations on 10% and 1% of current noise levels, and we were able to show that it would be possible to attain much higher fidelities. Furthermore, certain observables could be well represented with larger amounts of noise. Importantly, the simulation of an open system can be more accurate than unitary evolution as the open system dynamics could be closer to how a noisy computer is already affecting a state.

Future developments in noise reduction in the hardware, in post-processing error mitigation, as well as in reducing the number of gates for unitary evolutions can lead to a significant increase in simulation power.

In our system, we have limited the dissipation to the spins. An interesting avenue for future work could be the inclusion of loss in the bosonic degrees of freedom of the cavity, for which additional auxiliary qubits, gates, and connectivity requirements could prove challenging.

## Figures and Tables

**Figure 1 entropy-24-01766-f001:**
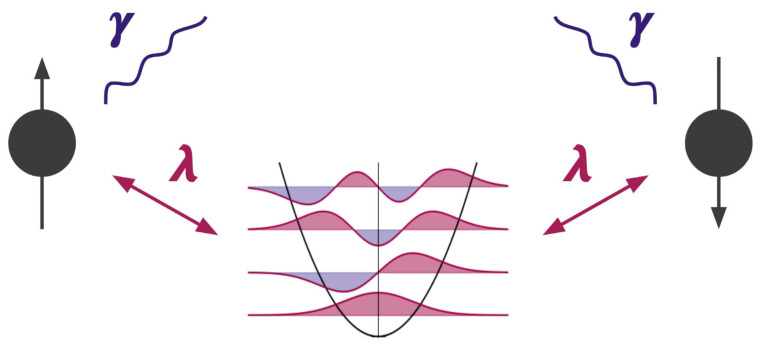
Depiction of the model described by Equations (Equation 1) and (Equation 2) for a number of spins $N_{S}=2$. The two spin sites are coupled to one harmonic oscillator of frequency ω via coupling parameter λ. Each of the spins dissipates independently into the environment at a rate of γ.

**Figure 2 entropy-24-01766-f002:**
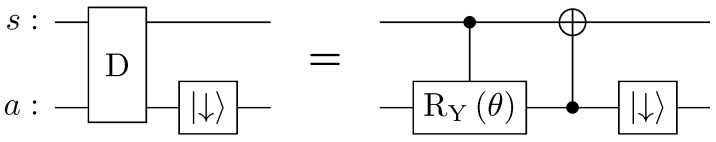
Circuit implementation of the dissipative part of the circuit, *D*, which represent a single collision to model Equation (Equation 2). *s* is the qubit representing the spin while *a* represents the auxiliary qubit.

**Figure 3 entropy-24-01766-f003:**
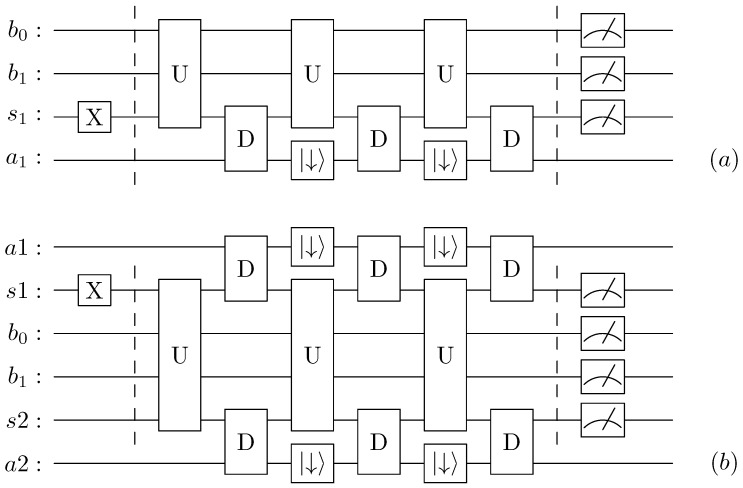
Circuit structure, alternating between a unitary evolution and collisions with auxiliary qubits and resets. (**a**) For a single spin and a (**b**) two spin system. Here, the spins are represented by $s_{k}$, the harmonic oscillator modes (4 levels) are encoded in the $b_{k}$ qubits and the auxiliary qubits are represented by $a_{k}$. X-Gate represents the initial state preparation, |↓〉 represents resets, the final gates represent measurements, while *D* is described in Figure 2.

**Figure 4 entropy-24-01766-f004:**
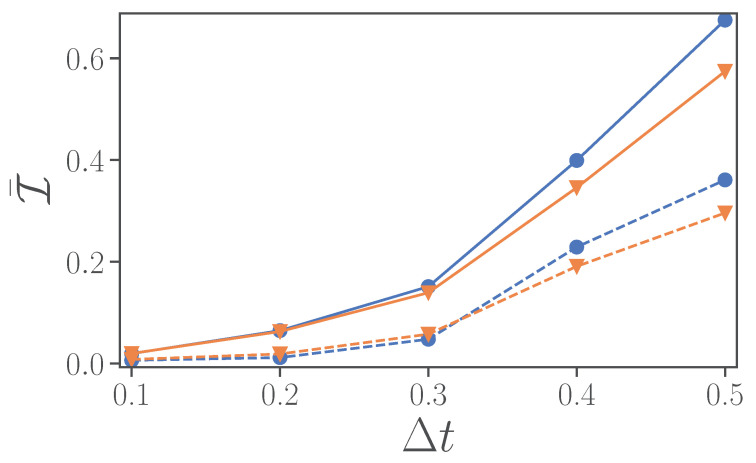
Time-averaged infidelity for the evolution from t=0 to t=2. Noiseless simulations of the Hamiltonian γ=0 (blue line, dots) and the open system γ=1 (orange line, triangle). The solid and dashed lines are used, respectively, for first-order and second-order Trotter implementations. The common parameters are ϵ=0.5, ω=4, λ=2. The number of time-steps for Δt=t/N=0.1,0.2,0.3,0.4,0.5 are N=20,10,7,5,4, respectively, not counting t=0, which just consists of the initial state preparation.

**Figure 5 entropy-24-01766-f005:**
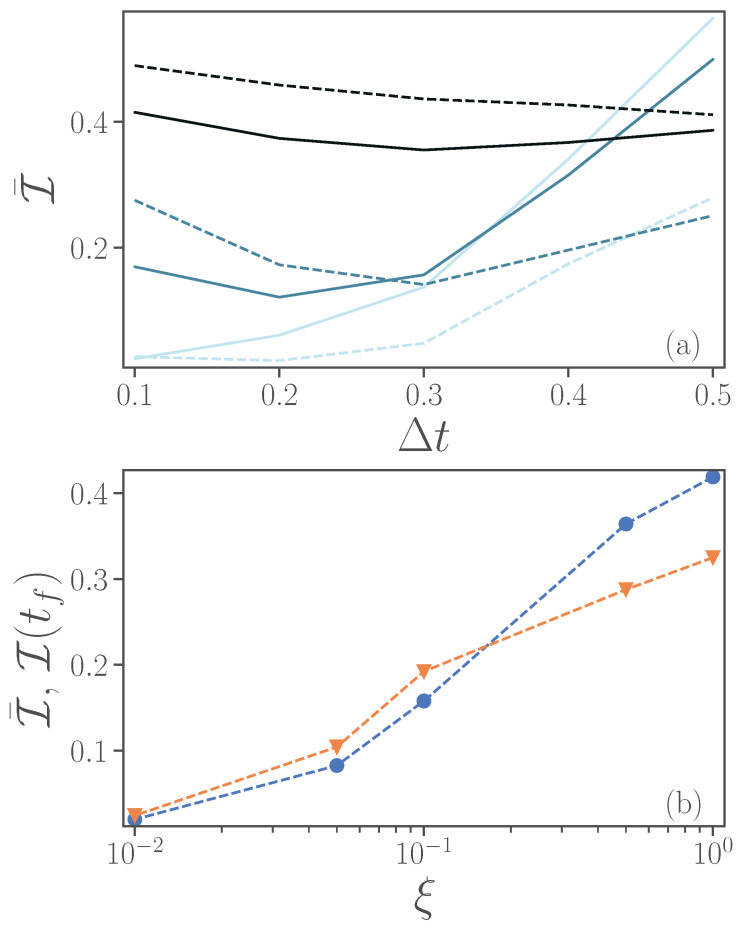
(**a**) Infidelity averaged over time as a function of time-step size Δt for an evolution from t=0 to a final time $t_{f}=2$. Different noise levels ξ=0.01,0.1,1 are represented by lighter to darker colors. (**b**) Time-averaged (blue circles) and final (orange triangles) infidelity as a function of noise levels. Here, the final time is taken as $t_{f}=2$ and we choose Δt=0.2. In both panels, results from first-order Trotter implementations are represented by continuous lines, while those from second-order are represented by dashed lines. Parameters ϵ=0.5, ω=4, λ=2, γ=1.

**Figure 6 entropy-24-01766-f006:**
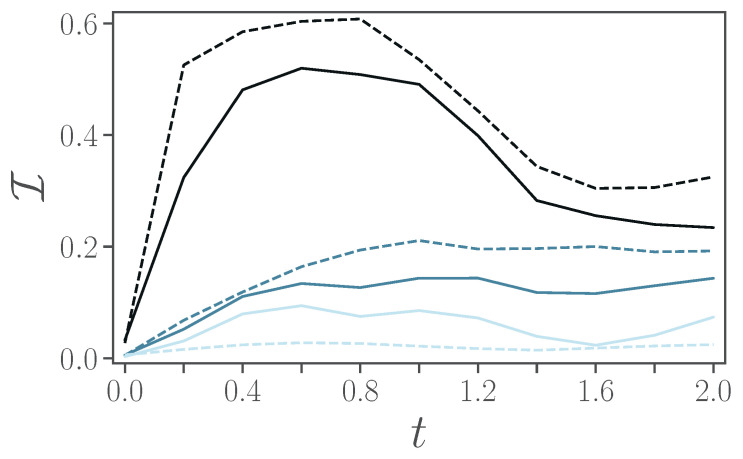
Infidelity as a function of time in open-system simulation in presence of noise. Using first-order Trotter (solid) and second-order Trotter (dashed) at Δt=0.2. At noise-factor ξ=0.01,0.1,1 (from lighter to darker colors). Other parameters are ϵ=0.5, ω=4, λ=2, γ=1.

**Figure 7 entropy-24-01766-f007:**
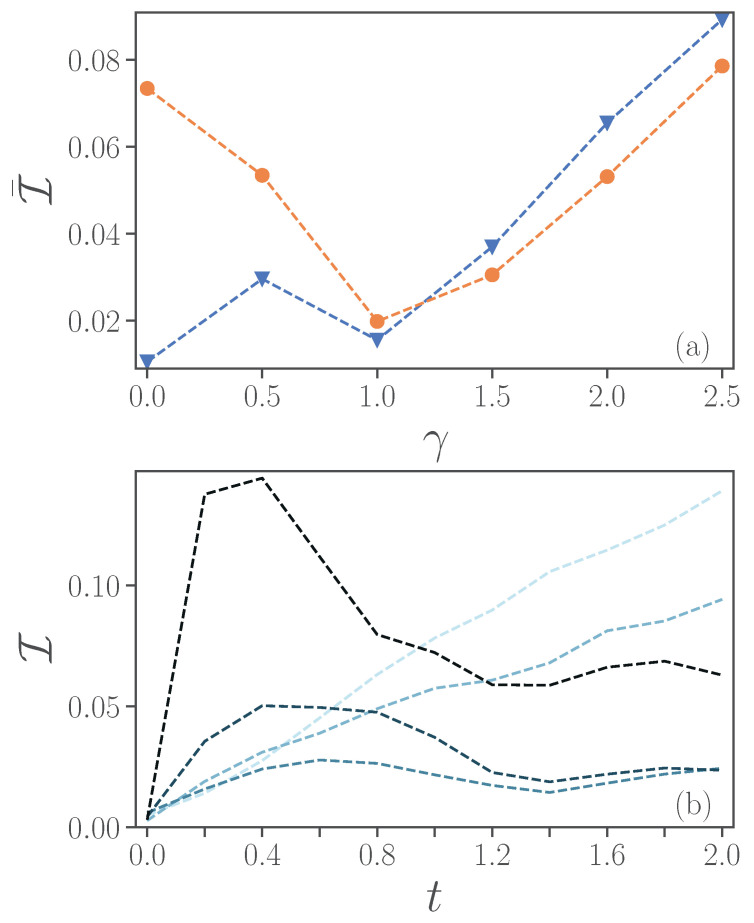
(**a**) Infidelity averaged over time versus dissipative rate γ with noise ξ=0.01 (orange line with circles) and without noise (blue line with triangles). (**b**) Infidelity as a function of time γ=0,0.5,1,1.5,2,2.5 (from lighter to darker colors). Second-order Trotter at Δt=0.2 and the other parameters are ϵ=0.5, ω=4, λ=2.

**Figure 8 entropy-24-01766-f008:**
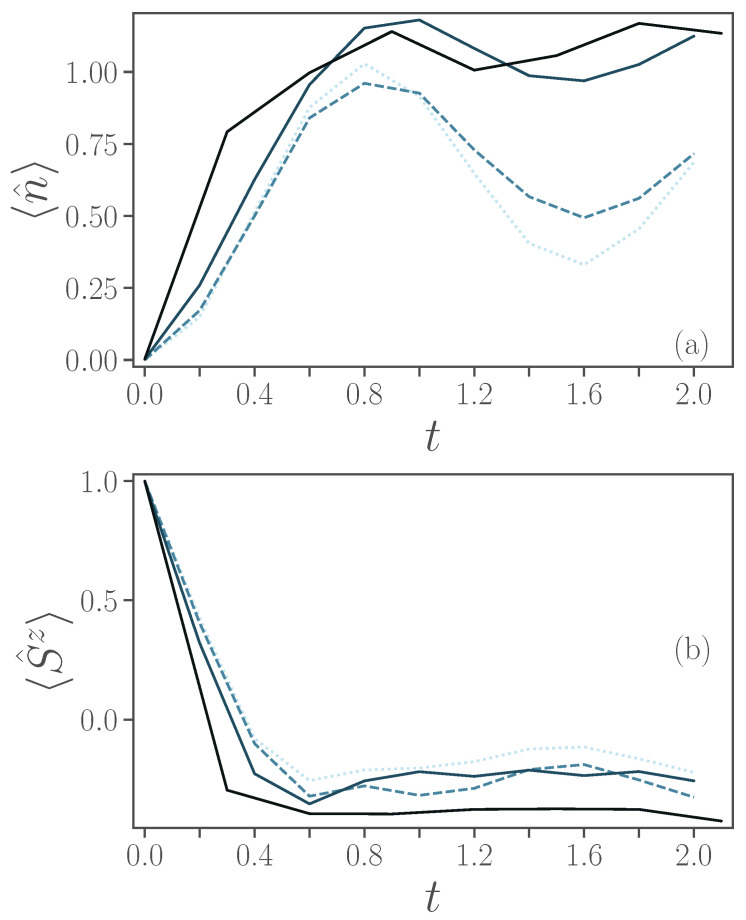
(**a**) Average bosonic occupation $\langle \hat{n}\rangle$ and (**b**) $\langle {\hat{\sigma }}^{z}\rangle$ as a function of time. Different noise levels ξ=0.01,0.1,1 are presented, respectively, by lighter to darker colors. As a reference, exact simulations are depicted by dotted lines. Results obtained using first-order Trotterization are with solid lines, while second-order with dashed lines. Other parameters are ϵ=0.5, ω=4, λ=2 and γ=1.

**Figure 9 entropy-24-01766-f009:**
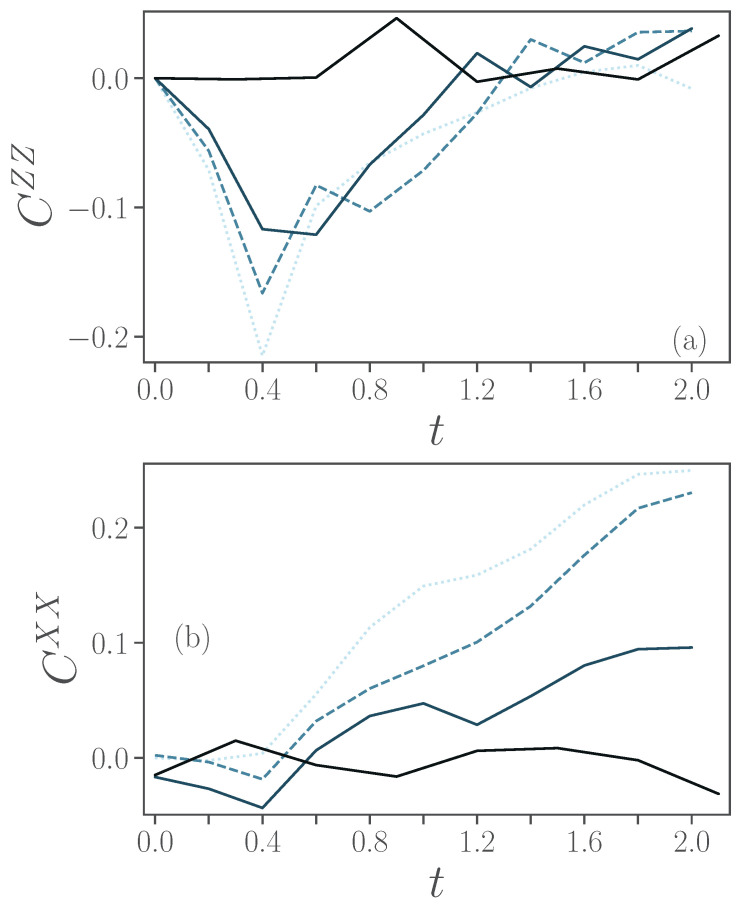
Correlations for the case of two spins: (**a**) spin-z connected correlation $C^{ZZ}$ (**b**) spin-x connected correlation $C^{XX}$ as a function of time. Different noise levels ξ=0.01,0.1,1 are presented, respectively, by lighter and darker colors. As a reference, exact simulations are depicted by the dotted lines. Results obtained using first-order Trotterization are represented by solid lines, while those using second-order Trotterization are represented by dashed lines. Other parameters are ϵ=0.5, ω=6, λ=2 and γ=1.

## Data Availability

The data presented in this study are available on request from the corresponding author.

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
