# Peer review of "Digital Quantum Simulation of the Spin-Boson Model under Markovian Open-System Dynamics"

_entropy, 2022, doi:10.3390/e24121766_

Round 1

Reviewer 1 Report

The work by A. Burger et al., entitled: “Digital quantum simulation of the spin-boson model under open systems dynamics” deals with the problem of implementing a simple example of an open quantum system in a quantum computer. The work could potentially be of interest to the community working on the field of quantum computing and quantum simulation; however, there are some points that I believe need to be addressed before considering the manuscript for publication:

1) I believe the titles is a bit misleading, the authors describe the simplest case of an open quantum system, namely Markovian evolution. I believe the title should be modified in order to reflect that the simulated system is a Markovian one.

2) In section 2, the authors start describing the system as Ns non-interacting spins coupled to a single harmonic oscillator, as well as to a bath. Then “bath” is then simplified by introducing a rather phenomenological dissipator, namely the Lindblad loss operator. Here I have two questions:

(i) What would be the reason for the losses to only affect the spin and not the harmonic oscillator? If there is a bath surrounding the system, one would expect that the whole system is affected by dissipation. Is there a physically realistic reason as to why one can only consider losses on the spin or is it only because that model is simpler to simulate?

(ii) I believe the notation needs to be corrected, because while in Eq. (2) the state of the spin is represented by arrows; in Appendix G, 0s and 1s are used instead. I guess the downwards arrow represents the vacuum state, whereas the upwards arrow represents the excited state of the spin.

3) In section 3.3 the authors introduce two correlators in Eq. (9). I believe, for the sake of the reader, that the authors should include a discussion on what these correlators represent or justify why these are a good measure of correlation between the two spins. It could be worth adding a reference where these correlators have previously been used.

4) In the discussion around Fig. 7, the authors mention that infidelity is lower for higher values of \gamma. However, they do not include any discussion around this interesting result. Why is the infidelity the lowest at \gamma = 1? Do you observe the same results for any combination of the system’s parameters? I believe that value is related to the coupling between the spin and the harmonic oscillator, and the fact that the spin is the only one coupled to the loss channel.

5) Finally, I believe it would be very helpful if the authors could share the algorithm (either as supplementary material or as a file in a public repository) to actually validate their results. The 7-qubit IBM-QPU is free to use, so the readers could eventually run/test the authors’ program.

Author Response

Please see the attachment for the point-by-point response and the revised paper.

Reviewer 2 Report

The authors studied open-system dynamics of a spin-boson model coupled to a dissipative channel on a digital quantum computation, i.e., IBM's Qiskit software. It is an interesting work but something should be clarified.

1. What is the range of noise-factor \xi? Maybe it has from 0 to 1?

2. [In 3.1, line 144] What does it mean that 'the unitary step is the main contribution to the infidelity' ? Please provide more explanation.

3. [In 3.2, line 156] What is the intermediate value of noise? Could you tell specific values?

4.  Fig.4 is as a function of \Delta t? Since \Delta t=t/N, what is the number N for each \Delta t?

5. [In line 171~172] It is not clear that the simulation of the quantum computer shows generally better performance. Better than what?

6. [In line 178~179] It is not clear that for larger values of \gamma, the infidelity can decrease after an initial time.  How much is the \gamma large? Instead of saying 'after an initial time', isn't it some point that the infidelity is maximized?

7. Is the Eq.(1) a typical Hamiltonian in a spin-boson model? Please put any references on it. 

Author Response

(The authors gave the same response as above.)

Round 2

Reviewer 1 Report

The authors have addressed all the points raised in my previous report. I recommend the manuscript for publication.

Minor comment:

Page 6 (right column): Remove the first paragraph (in blue font). The text corresponds to the review file!

Reviewer 2 Report

I am satisfied with the authors' reply.